# Continuous Matrix Product States for Inhomogeneous Quantum Field Theories: a Basis-Spline Approach

Martin Ganahl

*Perimeter Institute for Theoretical Physics, 31 Caroline Street North, Waterloo, ON N2L 2Y5, Canada*[*]

Continuous Matrix Product States (cMPS) are powerful variational ansätze for ground states of continuous quantum field theories in (1+1) dimension. In this paper we introduce a novel parametrization of cMPS wave functions using basis-splines, which we coin spline-based MPS (spMPS). We show how to variationally optimize spMPS by extending a recently developed ground state optimization algorithm for translational invariant cMPS [M. Ganahl, J. Rincón, G.Vidal. Phys.Rev.Lett. 118,220402 (2017)] to the case of spMPS, and use it to obtain the ground state of a gas of Lieb-Liniger bosons in a periodic potential. The resulting ground state wave function is virtually free of discretization artifacts, which we verify by comparing our results to lattice Density Matrix Renormalization Group (DMRG) calculations on extremely fine grids. We show that spMPS achieve a reduction of variational parameters by more than two orders of magnitude as compared to the lattice DMRG method, without any compromise in accuracy.

## I. INTRODUCTION

One of the big challenges in modern x1computational physics is the treatment of interacting quantum theories of many particles. Strong interaction between particles, such as electron-electron repulsion, electron-phonon interactions, or magnetic exchange interactions between spin degrees of freedom, are considered to be vital to phenomena like high temperature superconductivity[1] or the fractional quantum Hall effect[2,3]. In most cases, solving the relevant models exactly is beyond current capabilities, and a big effort has thus been devoted to develop approximate analytic and numerical methods to tackle these systems. For one dimensional (1d) quantum systems, so called Matrix Product States[4,5], have proven to be particularly powerful tools to obtain ground states[5] and also low-lying excited states[6–8] of quantum lattice systems. In particular, the Density Matrix Renormalization Group[5,9] (DMRG), a ground state optimization method within the variational class of MPS, is today considered the most powerful method for obtaining ground states of 1d quantum lattice systems. Inspired by the successes for lattice theories, growing effort has been devoted to adapt the lattice DMRG method to continuous quantum field theories[10–14], using lattice discretization techniques.

In a parallel development, alternative tensor network approaches have been proposed which work directly in the continuum[15,16]. Continuous Matrix Product States (cMPS), a prominent example of such a continuous tensor network, have been shown to be well suited to obtain ground states of continuous quantum field theories directly in the continuum[17–26], without the need to introduce lattice discretizations. These methods have interesting applications ranging from cold atomic gases[27,28] to quantum chemistry to applications even in holography and quantum gravity[29,30]. For translational invariant theories, efficient methods have been developed to obtain accurate ground state wave functions for infinite systems[26,31] or systems on a ring[25], as well as low-lying excited states[18]. For the case of inhomogeneous Hamiltonians, first important results have been obtained[22]. However, despite a considerable ongoing effort, no successful optimization for inhomogeneous cMPS has been achieved so far. The present manuscript aims at filling this gap. The main contribution of this paper is the introduction of a spline-based representation for inhomogeneous cMPS which we coin spline-based Matrix Product States (spMPS), and the development of computational optimization tools for spMPS. In the following, we will discuss in depth the computational advantages of this representation, and will develop new tools for manipulating and optimizing spMPS. We will demonstrate the power of the spMPS ansatz by obtaining the ground state of a gas of interacting Lieb-Liniger bosons in a periodic potential.

## II. CONTINUOUS MATRIX PRODUCT STATES AND BASIS-SPLINE INTERPOLATION

In the following we will consider a gas of a single species of bosons on the real line, with a periodic unit-cell of length $L = 1$. For such a system, a generic cMPS wave function assumes the form

$$|\Psi\rangle = v_l \mathcal{P} e^{\int_{-\infty}^{\infty} dx \, Q(x)\otimes\mathbb{1}+R(x)\otimes\psi^\dagger(x)} v_r |0\rangle, \quad (1)$$

where $Q(x), R(x) \in \mathbb{C}^{D\times D}$ are periodic matrix functions with period $L$, and $\psi^\dagger(x)$ is a bosonic creation operator for the vacuum state $|0\rangle$. $v_l$ and $v_r$ are arbitrary boundary vectors at $x = \pm\infty$, and $\mathcal{P}$ is the path-ordering operator. For given $Q(x), R(x)$, any local observable $\langle\Psi|O(x)|\Psi\rangle$, e.g. $O(x) = \frac{d\psi^\dagger(x)}{dx}\frac{d\psi(x)}{dx}$, can be calculated using contraction techniques similar to the lattice case. In particular, the expectation value of energy *densities* can be evaluated. This enables the application of the variational principle to the class of cMPS wave functions, and for homogeneous systems has lead to the development of efficient methods for calculating ground state wave functions of continuous quantum field theories[26,31].

In numerical applications, the functions $Q(x)$ and $R(x)$ are usually not known analytically.

Instead of using continuous functions $Q(x), R(x)$, a common approach in numerical mathematics is to define a set of points $(x_i, Q(x_i))$ and $(x_i, R(x_i))$ on a fine grid $x_i, i = 1 \ldots M_{grid}$, and use these points to approximate continuous functions $Q(x), R(x)$. When aiming to approximate the ground state of a continuous Hamiltonian $H$, one then uses the same discretization for the Hamiltonian. The number $M_{grid}$ of grid points should be chosen according to the smallest scale of variation in any of the Hamiltonian parameters. For example, if the Hamiltonian contains a chemical potential of cosine form with a period of one, $\mu(x) \sim \cos(2\pi x)$, one might expect the same period (and possibly higher harmonics) to be present in the ground state wave function. Thus, $M_{grid}$ has to be chosen so large that at least this variation may be well resolved. This discretization, however, introduces an error in the evaluation of expectation values (see discussion below). In this paper, we propose a new parametrization of the cMPS wave function and combine it with a recently proposed optimization method for homogeneous cMPS. Our central proposal is to use basis-spline (b-spline in the following) interpolation to parametrize the continuous functions $Q(x), R(x)$. For a detailed introduction to b-splines we refer the reader to the excellent review by Bachau et al.[32]. In the following, we will give a short introduction to b-splines and summarize their most important properties.

### A. An introduction to basis-splines

Given a set of $P$ discrete data points $(x_i, f_i)$ with $x_i, f_i \in \mathbb{R}$, and $x_1 < x_2 \cdots < x_P$, a frequently encountered task in numerical analysis is to find a smooth curve $\tilde{f}(x)$ which runs through all points $(x_i, f_i)$, i.e $\tilde{f}(x_i) = f_i$. Smooth means that the curve should be differentiable up to some order $p$ everywhere inside the domain $x_1 < x < x_P$, with the possible exception of the boundary points $x_1, x_P$. The simple yet powerful idea behind b-spline interpolation is to write $\tilde{f}(x)$ as a piece-wise polynomial function. Piece-wise polynomial here means that $\tilde{f}(x)$ has an expansion in a set of polynomials $B_i^k(x)$ of degree $k$,

$$\tilde{f}(x) = \sum_{i=1} \beta^i B_i^k(x), x \in [x_1, x_P], \qquad (2)$$

where every $B_i^k(x)$ is non-zero only inside a small region, and zero everywhere else, and $\beta^i$ are expansion coefficients. The $B_i^k(x)$ are designed to be sufficiently often differentiable, such that $\tilde{f}(x)$ has the desired smoothness properties. The coefficients $\beta^i$ are chosen such that $\tilde{f}(x_i) = f_i$. Fig. 1 (a) shows an example of a simple b-spline interpolation. We chose thirteen equally spaced points $x_i \in [0, 1]$ and generated the data points

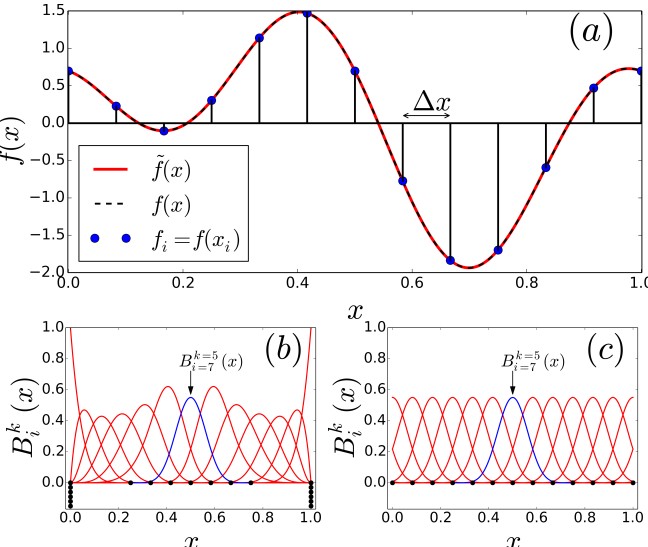

FIG. 1: **Example of b-spline interpolation** (a) We generate data-points $(x_i, f_i = f(x_i))$ (blue dots) using the values $f(x_i)$ of Eq.(3) at 13 equally spaced points $x_i \in [0, 1]$ with spacing $\Delta x$. We then use a b-spline interpolation to obtain the function $\tilde{f}(x)$ (red solid line). For comparison we also show $f(x)$ (black dashed line). (b) B-spline polynomials $B_i^k(x)$ for spline interpolation with open boundary conditions. (c) B-spline polynomials $B_i^k(x)$ for spline interpolation with periodic boundary condition. We have highlighted the polynomial $B_7^5(x)$ in blue.

$(x_i, f_i = f(x_i))$ from the function

$$f(x) = \sin(2\pi x) + \cos(4\pi x + 0.8). \qquad (3)$$

We then used a degree $k = 5$ interpolation to obtain the function $\tilde{f}(x)$. All three are plotted in Fig. 1(a). In particular, the function $\tilde{f}(x)$ is seen to approximate $f(x)$ very well, with a relative error less than 0.1%. In Fig. 1(b) we plot the corresponding polynomials $B_i^k(x)$, and, additionally, the so called knot-points $t_j$[32] as black dots. The $\{t_j\}$ are a set of ascending, not necessarily distinct points with $x_1 \leq t_j \leq x_P$. If a point $t_j$ appears multiple $(\nu_j)$ times, it is said to have multiplicity $\nu_j$. The knot points $t_j$ and their multiplicities $\nu_j$ are part of the definition of the $B_i^k(x)$. The $t_j$ are determined from the points $x_i$, and for standard applications, their multiplicity is chosen to be $\nu_1 = \nu_P = k + 1$ and $\nu_j = 1$ for $j = 2 \ldots P - 1$. We have indicated the multiplicity of the points $t_j$ in Fig. 1(b) accordingly. As can be seen from Fig. 1(b), a polynomial $B_i^k(x)$ has support on the interval $]t_i, t_{i+1}[$, and vanishes outside of it. This is illustrated for $B_7^5(x)$, highlighted in blue. Note that for $0 < x < 1$, all polynomials, together with their derivatives $\partial^n B_i^k(x), n = 0, \ldots, k - 1$ are continuous (denoted by $B_i^k \in C^{k-1}$). At $x = 0$, however, the first $k + 1$ polynomials are $B_i^k(x) \in C^{i-1}$, for $i = 1, \ldots k + 1$ (and likewise for $x = 1$). For example, $B_0^5(x)$ is discontinuous at $x = 0$, $B_1^5(x)$ is continuous, but has a discontinuous first derivative, a.s.o (and likewise $B_{13}^5(x), B_{12}^5(x), \ldots$ at

$x = 1$).

So far we did not make use of the fact that $f(x)$ in Eq.(3) is periodic. The b-spline polynomials $B_i^k(x)$ for a periodic expansion are shown in Fig. 1(c). The knot points are in this case equidistantly spaced and chosen to have multiplicity one, and all $B_i^k(x)$ differ only by a shift in $x$ and are $B_i^k(x) \in C^{k-1}$. Using periodic b-spline interpolation has the advantage of giving a higher accuracy when approximating a periodic function $f(x)$ with $\tilde{f}(x)$. In this paper, we will thus use periodic b-spline interpolation throughout. Most packages for numerical computation provide efficient routines to generate and handle b-spline interpolations. For this work, we use the routines *splrep* and *splev* provided by the *scientific python*[33] package.

### B. Spline-based Matrix Product States

Returning to our previous discussion of cMPS, we propose to parametrize the continuous matrix functions $Q(x), R(x)$ as

$$Q_{\alpha\beta}(x) = \sum_i \mathcal{Q}_{\alpha\beta}^i B_i^k(x)$$

$$R_{\alpha\beta}(x) = \sum_i \mathcal{R}_{\alpha\beta}^i B_i^k(x), x \in [0, L].$$

The expansion coefficients $\mathcal{Q}_{\alpha\beta}^i, \mathcal{R}_{\alpha\beta}^i$ contain the variational parameters of our ansatz state, and $\{B_i^k(x)\}$ are a given set of b-spline polynomials on an interval $I = [0, L]$. Such a parametrization has several appealing features. For example, as we have illustrated above, any sufficiently smooth function can be very accurately approximated by a small number of basis spline functions. If the matrices $Q(x)$ and $R(x)$ are expected to be sufficiently smooth, this parametrization is more efficient than storing matrices $Q(x), R(x)$ on a very fine grid with $M_{grid}$ grid points. Another appealing property is that the matrices $Q(x), R(x)$ can be evaluated at any point $x \in I$. This is a significant advantage for example when considering the action of local operators on the cMPS $|\Psi\rangle$, as required in e.g. minimization algorithms. For example, in the cMPS framework, the action of the operator $\frac{d\psi(x)}{dx} |\Psi\rangle$ is translated into the operation $\frac{d\psi(x)}{dx} |\Psi\rangle \sim [Q(x), R(x)] + \frac{dR}{dx}$ on the matrices $Q(x), R(x)$[34]. The derivative $\frac{dR}{dx}$ can be obtained analytically from the spline representation of the matrices. Compared to a finite-difference approximation $\frac{dR}{dx} \approx \frac{R(x_i) - R(x_{i-1})}{x_i - x_{i-1}}$ on a discrete set of points $(x_i, R(x_i))$, the former approach has a higher accuracy. A major benefit of the b-spline parametrization is thus a strong reduction in discretization errors, as compared to a standard discretization methods using lattice MPS.

One of the main differences of our proposed approach as compared to regular lattice-MPS discretization methods is that in the latter case one introduces a discretization of the Hamiltonian with a finite lattice spacing $a$,

and an identical discretization for the wave function $|\Psi\rangle$. For the simplest discretization scheme, the error of observables is of order $a$ throughout the calculation (higher order discretization of derivatives usually yield more accurate results, at the cost of introducing longer ranged terms in the discretized Hamiltonian). In our proposed approach on the other hand, the error is determined by how well the spline-interpolation can capture the typical oscillations present in a periodic cMPS. For ground states, it is reasonable to expect that these oscillations are of the same order as those present in the Hamiltonian parameters. For example, if these oscillations are of the same order as those shown in Fig. 1(a), a small number of b-spline polynomials will already suffice to give very accurate results. In particular, the accuracy will be much higher than the naive lattice spacing $\Delta x$ used in the figure. This is evident from the high accuracy with which the function $f(x)$ from Eq.(3) is reproduced by the b-spline interpolation $\tilde{f}(x)$ Eq.(2).

## III. REGAUGING AN INHOMOGENEOUS CMPS IN THE THERMODYNAMIC LIMIT

Before proceeding, it is useful to introduce some diagrammatic notation at this point and elaborate on its relation to regular lattice MPS diagrams. For a gas of a single species of bosons (which we will be dealing with in this paper), the cMPS can be thought of as a collection of matrices $A^\sigma(x)$ of the form

$$A^0(x) = \mathbb{1} + \epsilon Q(x), \sigma = 0$$

$$A^\sigma(x) = \sqrt{\frac{\epsilon^\sigma}{\sigma!}} [R(x)]^\sigma, \sigma > 0$$

at any point $x$ in space. Here $\epsilon$ is an arbitrarily small discretization parameter which will be sent to 0 at the end of all calculations, i.e. $\epsilon \to 0$. For the cases considered in this manuscript, tensors $A^\sigma$ with $\sigma > 1$ can be neglected. They contribute to the energy density with a subleading order in $\epsilon$, and will thus have a vanishing contribution in the limit $\epsilon \to 0$. The assumption is that contributions to the wave function with higher and higher number of particles are exponentially supressed. We will thus use the following shorthand notations interchangeably:

$$A^\sigma(x) = \begin{matrix} \boxed{A(x)} \end{matrix} = \begin{pmatrix} 1 + \epsilon Q(x) \\ \sqrt{\epsilon} R(x) \end{pmatrix} = \begin{matrix} \boxed{\begin{smallmatrix} 1 + \epsilon Q(x) \\ \sqrt{\epsilon} R(x) \end{smallmatrix}} \end{matrix}$$

where we have suppressed contributions with $\sigma > 1$. In the translational invariant case $A^\sigma(x) = A^\sigma = const.$, the cMPS can be drawn as

$$|\Psi\rangle = \ldots \boxed{A} \boxed{A} \ldots .$$

## A.  Regauging techniques for cMPS

Two objects of central importance in any MPS calculation in the thermodynamic limit are the left and right reduced steady-state density matrices $\langle l(x)|, |r(x)\rangle \in \mathbb{C}^{D\times D}$. They are defined as the eigenmatrices with eigenvalues $\eta = 1$ of the so-called unit-cell transfer operator $\mathcal{T}(0,L)$

$$\mathcal{T}(0,L) = \mathcal{P}e^{\int_0^L T(x)dx}$$
$$T(x) = Q(x)\otimes\mathbb{1} + \mathbb{1}\otimes Q^*(x) + R(x)\otimes R^*(x). \quad (4)$$

A star $^*$ denotes complex conjugation. In formulas, $\langle l(x)|$ and $|r(x)\rangle$ are given by

$$\langle l(0)|\,\mathcal{T}(0,L) = \langle l(0)| \quad (5)$$
$$\mathcal{T}(0,L)\,|r(L)\rangle = |r(L)\rangle \quad (6)$$
$$\langle l(x)| = \langle l(0)|\,\mathcal{T}(0,x) \quad (7)$$
$$|r(x)\rangle = \mathcal{T}(x,L)\,|r(L)\rangle. \quad (8)$$

The bra-ket notation here should emphasize the vector-character of $\langle l(x)|$ and $|r(x)\rangle$. We use the following diagrams to represent generic matrices $\langle f(x)|$ and $|g(x)\rangle$ of this character:

$$f(x) = \langle f(x)| = \boxed{f(x)}$$

$$g(x) = |g(x)\rangle = \boxed{g(x)} \ .$$

The cMPS transfer operator $T(x)$ of Eq.(4) acts as the derivative operator of $|r(x)\rangle, \langle l(x)|$,

$$\frac{d\,|r(x)\rangle}{dx} = T\,|r(x)\rangle \quad (9)$$
$$= Q(x)r(x) + r(x)Q^\dagger(x) + R(x)r(x)R^\dagger(x)$$
$$\frac{d\,\langle l(x)|}{dx} = \langle l(x)|\,T \quad (10)$$
$$= l(x)Q(x) + Q^\dagger(x)l(x) + R^\dagger(x)l(x)R(x).$$

To obtain $\langle l(x)|$ (or $|r(x)\rangle$) for a given $\langle l(0)|$ (or $|r(L)\rangle$) at any point $x > 0$ inside the unit-cell one has to solve the boundary value problem for the system of ordinary differential equations (ODEs) of Eq.(9) and Eq.(10). In this paper we use a fifth order Dormand-Prince[35] routine as provided by the *scientific python* package (dopri5) to integrate these equations. The dopri5 routine (as many other high-accuracy solvers for ODEs) chooses the step size in the integration according to a specified error bound. Hence, it is crucial that the differential operator $T(x)$ can be evaluated at arbitrary points $x \in I$. Using the above introduced diagrams, the evolution of $\langle l(0)|$ over the unit-cell can be drawn as an infinite tensor network of the form

$$\langle l(0)|\,\mathcal{T}(0,L) = \boxed{l(0)}\ \boxed{A(0)}\ \boxed{A(\epsilon)} \cdots \boxed{A(L)} \ .$$

One of the main obstacles in cMPS optimization is the lack of algorithms to change the so called *gauge*[34] of a generic cMPS. Algorithms for optimization of lattice MPS make frequent use of such regauging techniques[36,37] in order to improve stability and speed up convergence. For the simplest case of a translational invariant (c)MPS $|\Psi\rangle$ with tensors $A^\sigma$, a gauge transformation is a similarity transformation

$$A^\sigma \leftarrow X A^\sigma X^{-1}$$

with an arbitrary invertible matrix $X$. Such a gauge transformation can be used to bring the (c)MPS tensors into left or right canonical form[38]

$$\boxed{\begin{matrix} A \\ A \end{matrix}} = \Big( \qquad \text{(left)}$$
$$\boxed{\begin{matrix} A \\ A \end{matrix}} = \Big) \quad \text{(right).}$$

$\Big)$ and $\Big($ are graphical notation for identity operators. For non-translational invariant lattice MPS, regauging techniques make frequent use of singular value (SV) and QR decomposition. For non-translational invariant cMPS on the other hand, SV and QR decomposition have not yet been developed, and hence no methods for regauging such a cMPS are available so far. In the following we propose a method which fills this gap and allows for an arbitrary regauging of a periodic cMPS in the thermodynamic limit. Let us first recall the definition of left and right orthogonality of a cMPS[34]. Similar to the case of lattice MPS[38], cMPS matrices $Q_l(x), R_l(x)$ or $Q_r(x), R_r(x)$ are said to be in left or right orthogonal form at position $x$ if they obey

$$\langle\mathbb{1}|\,T_l(x) = Q_l(x) + Q_l^\dagger(x) + R_l^\dagger(x)R_l(x) = 0 \quad (11)$$
$$T_r(x)\,|\mathbb{1}\rangle = Q_r(x) + Q_r^\dagger(x) + R_r(x)R_r^\dagger(x) = 0. \quad (12)$$

The methods we are presenting in the following are generalizations of normalization and regauging procedures for periodic lattice MPS to the case of cMPS.

**Normalization of a cMPS:** As the first step, we determine the steady-state reduced density matrices $\langle l(0)|, |r(L)\rangle$ by finding the dominant left and right eigenmatrices of $\mathcal{T}(0,L)$ (using for example a sparse eigensolver, in combination with the dopri5 routine):

$$\langle l(0)|\,\mathcal{T}(0,L) = \eta\,\langle l(0)|$$
$$\mathcal{T}(0,L)\,|r(L)\rangle = \eta\,|r(L)\rangle$$

with $\eta \in \mathbb{R}$. The state is normalized if $\eta = 1$. For $\eta \neq 1$, we can normalize it by the transformation

$$Q(x) \leftarrow Q(x) - \frac{\ln\eta}{2L}\mathbb{1}$$

at any position $x$.

**Left canonical form of a cMPS:** From $\langle l(0)|$ we then obtain $\langle l(x)|$ from Eq.(7), which is used to transform the cMPS matrices into left orthogonal form by the gauge transformation

$$\begin{pmatrix} 1 + \epsilon Q(x) \\ \sqrt{\epsilon} R(x) \end{pmatrix} \to \sqrt{l(x)} \begin{pmatrix} 1 + \epsilon Q(x) \\ \sqrt{\epsilon} R(x) \end{pmatrix} \left[ \sqrt{l(x+\epsilon)} \right]^{-1} \tag{13}$$

where $\sqrt{l(x)}$ is the matrix square root of $l(x)$. We have simply inserted an identity $\mathbb{1} = \sqrt{l(x)}[\sqrt{l(x)}]^{-1}$ on each link between the tensors $\begin{pmatrix} 1 + \epsilon Q(x) \\ \sqrt{\epsilon} R(x) \end{pmatrix}$. We now Taylor-expand $[\sqrt{l(x+\epsilon)}]^{-1}$ in order to express Eq.(13) with objects at position $x$. Expanding $\sqrt{l(x+\epsilon)}$ into

$$\sqrt{l(x+\epsilon)} = \sqrt{l(x)} + \epsilon \frac{d\sqrt{l(x)}}{dx}$$
$$= \Big(1 + \epsilon \underbrace{\frac{d\sqrt{l(x)}}{dx}\left[\sqrt{l(x)}\right]^{-1}}_{\phi(x)}\Big)\sqrt{l(x)}$$

we obtain

$$\left[\sqrt{l(x+\epsilon)}\right]^{-1} = \left[\sqrt{l(x)}\right]^{-1}\left(1 - \epsilon\phi(x)\right) + \mathcal{O}(\epsilon^2).$$

If we insert this back into Eq.(13) we get the result

$$\sqrt{l(x)}\begin{pmatrix} 1 + \epsilon Q(x) \\ \sqrt{\epsilon} R(x) \end{pmatrix}\left[\sqrt{l(x+\epsilon)}\right]^{-1} =$$
$$\begin{pmatrix} 1 + \epsilon\big(\sqrt{l(x)}Q(x)\left[\sqrt{l(x)}\right]^{-1} - \phi(x)\big) \\ \sqrt{\epsilon}\sqrt{l(x)}R(x)\left[\sqrt{l(x)}\right]^{-1} \end{pmatrix}$$

from which we can read off the transformation that takes $Q(x), R(x)$ into their left-orthogonal form:

$$Q_l(x) = \sqrt{l(x)}Q(x)\left[\sqrt{l(x)}\right]^{-1} - \frac{d\sqrt{l(x)}}{dx}\left[\sqrt{l(x)}\right]^{-1} \tag{14}$$

$$R_l(x) = \sqrt{l(x)}R(x)\left[\sqrt{l(x)}\right]^{-1}. \tag{15}$$

A similar procedure can be used to obtain the right canonical form of a cMPS.

**Central canonical form of a cMPS:** The above techniques can also be used to obtain what we call the *central canonical form* of an inhomogeneous cMPS. We will only state the result here and refer the reader to the appendix for a detailed calculation. The central canonical form of an inhomogeneous cMPS is given by matrix functions $\Gamma_Q(x), \Gamma_R(x), C(x)$ and $\frac{dC(x)}{dx}$ such that the left and right orthogonal matrices $Q_l(x), R_l(x)$ and

$Q_r(x), R_r(x)$ are obtained from

$$Q_l(x) = C(x)\Gamma_Q(x)$$
$$R_l(x) = C(x)\Gamma_R(x)$$
$$Q_r(x) = \Gamma_Q(x)C(x) + [C(x)]^{-1}\frac{dC}{dx}$$
$$R_r(x) = \Gamma_R(x)C(x).$$

It is a simple matter to verify that the matrices

$$\Gamma_Q(x) = \frac{1}{\sqrt{r(x)}}Q(x)\frac{1}{\sqrt{l(x)}}$$
$$\qquad - \frac{1}{\sqrt{r(x)}}\frac{1}{\sqrt{l(x)}}\frac{d\sqrt{l(x)}}{dx}\frac{1}{\sqrt{l(x)}} \tag{16}$$

$$\Gamma_R(x) = \frac{1}{\sqrt{r(x)}}R(x)\frac{1}{\sqrt{l(x)}} \tag{17}$$

$$C(x) = \sqrt{l(x)}\sqrt{r(x)} \tag{18}$$

will yield the desired result (see Eq.(21) for a diagrammatic representation of a cMPS in central canonical form). Note that unlike the central canonical form for homogeneous cMPS, we do not use an SVD to decompose $C(x)$ into its singular values.

The center, left and right orthogonal tensors $Q_C(x), R_C(x), Q_l(x), R_l(x)$ and $Q_r(x), R_r(x)$ are related by the transformation

$$Q_C(x) \equiv Q_l(x)C(x)$$
$$C(x)Q_r(x) = Q_C(x) + \frac{dC}{dx}$$
$$R_C(x) \equiv R_l(x)C(x) = C(x)R_r(x)$$

### B. Regauging for spMPS

So far the discussion has been general, without any reference to a particular representation of a cMPS. For the case of spMPS, the transformations can be implemented by making use of b-spline interpolations. The normalization of a spMPS is carried out by choosing a set of (not necessarily) equidistantly spaced points $x_i \in [0, L]$ and transforming the matrices $Q(x_i)$ at these points according to

$$Q(x_i) \leftarrow Q(x_i) - \frac{\ln \eta}{2L}\mathbb{1}.$$

A normalized spMPS is obtained from an interpolation of the points $(x_i, Q(x_i))$ (note that $R(x)$ remains unchanged). In the following we use $x_i$ with subscript $i = 1, \ldots, P$ to label the interpolation points of the cMPS matrices. To implement the regauging, we obtain a numerical solution $l(x_n)$ of Eq.(7) on a discrete set of points $x_n \in [0, L], n = 1 \ldots N$ (we use $x_n$ with subscript $n = 1, \ldots N$ to denote interpolation points of $l(x_n), r(x_n)$ or other objects of the same character). A

smooth function $l(x)$ is then again obtained from interpolating the points $(x_n, l(x_n))$, as described above. A similar approach is applied to obtain a smooth function $\sqrt{l(x)}$. The quality of the left and right orthogonality depends on the number of grid points $N$ and the order $k$ used to obtain the b-spline functions $l(x)$ and $\sqrt{l(x)}$. For the cases considered in this paper, we empirically found that using $P \approx 50$, $N = 200$ points $x_n$ and interpolating polynomials of order $k = 5$, an accuracy $\|\langle \mathbb{1}| T_l(x)\| < \mathcal{O}(10^{-8})$ (and similar for right orthogonalization) is easily achievable.

## IV. OPTIMIZATION OF SPMPS

In this section we will detail a method to obtain an approximate ground state of a quantum field theory that possesses a non-trivial unit-cell periodicity $L$. We will first summarize the basic strategy and then move on to explain the individual steps in more detail.

1. Given a spMPS $Q(x), R(x)$, choose a set of points $\{x_i\}$.

2. Regauge the state into the central canonical form with respect to the points $\{x_i\}$, i.e. calculate $Q_C(x_i), R_C(x_i), Q_l(x_i), R_l(x_i), Q_r(x_i), R_r(x_i), C(x_i)$ (and possibly their derivatives, see below).

3. Calculate an update $V(x_i), W(x_i)$ to the matrices $Q_C(x_i), R_C(x_i)$ (see below),

$$Q_C(x_i) \rightarrow \tilde{Q}(x_i) = Q_C(x_i) - \alpha V(x_i)$$
$$R_C(x_i) \rightarrow \tilde{R}(x_i) = R_C(x_i) - \alpha W(x_i).$$

This is expected to lower the energy of the state. $\alpha$ is a suitably chosen small, real number.

4. Obtain the matrices

$$Q(x_i) = \tilde{Q}(x_i)[C(x_i)]^{-1},$$
$$R(x_i) = \tilde{R}(x_i)[C(x_i)]^{-1}$$

5. Use a b-spline interpolation on the points $(x_i, Q(x_i)), (x_i, R(x_i))$ to obtain a new spMPS $Q(x), R(x)$ and go back to 2.

We will demonstrate in the following our proposed method for a non-integrable gas of Lieb-Liniger[39,40] bosons in a periodic potential. The Hamiltonian is given by

$$H = \int dx \, h(x) \equiv$$
$$\int dx \left( \frac{1}{2m} \partial_x \psi^\dagger(x) \partial_x \psi(x) + \mu(x) \psi^\dagger(x) \psi(x) \right.$$
$$\left. + g \, \psi^\dagger(x) \psi^\dagger(x) \psi(x) \psi(x) \right), \quad (19)$$
$$\mu(x) = \mu_0 \left( \cos(\frac{2\pi x}{L}) - 1 \right)^2 - \frac{1}{2}. \quad (20)$$

The algorithm sketched above is a generalization of the algorithm presented in[26] to the case of a non-homogeneous cMPS. We will here briefly recall the main steps of the method proposed in[26]. In the translational invariant case, the cMPS can be parametrized by two constant matrices $Q, R$, which can be gauged into left-orthogonal form $Q_l, R_l$ (see Eq.(11)). Pictorially, the state is given by

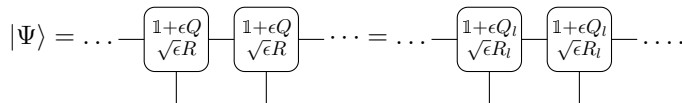

The optimization for the homogeneous case proceeds by gauging the state into the central canonical form

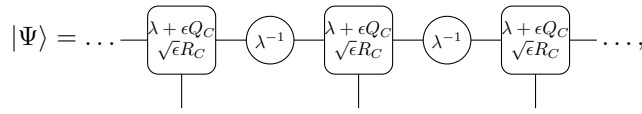

were $\lambda$ is a diagonal matrix containing the Schmidt values, and

$$Q_C = Q_l \lambda$$
$$R_C = R_l \lambda.$$

A local update $V, W$ to the matrices $Q_C, R_C$ is then calculated from the gradient of the energy,

$$V, W = \underset{Q_C, R_C}{\operatorname{argmin}} \frac{\langle \Psi | H | \Psi \rangle}{\langle \Psi | \Psi \rangle},$$

(see appendix in[26]) producing a new set of tensors

$$\tilde{Q} = (Q_C - \alpha V) \lambda^{-1}$$
$$\tilde{R} = (R_C - \alpha W) \lambda^{-1}$$

from which the above described procedure can be restarted ($\alpha$ is a small, real number).

Our generalization of this method to an inhomogeneous setting starts by choosing a set of (not necessarily) equidistantly spaced points $x_i \in [0, L], i = 1 \ldots P$, and bringing a spMPS into the central canonical form around these points. Pictorially, the state decomposition is given by

$$\cdots \boxed{\begin{array}{c} \mathbb{1} + \epsilon Q_l(x_i) \\ \sqrt{\epsilon} R_l(x_i) \end{array}} - C(x_i + \epsilon) - C^{-1}(x_i + \epsilon) \cdots \boxed{\begin{array}{c} \mathbb{1} + \epsilon Q_l(x_{i+1}) \\ \sqrt{\epsilon} R_l(x_{i+1}) \end{array}} - C(x_{i+1} + \epsilon) - C^{-1}(x_{i+1} + \epsilon) \cdots$$

which to order $\mathcal{O}(\epsilon^{3/2})$ is equivalent to

$$\ldots \boxed{\begin{array}{c} C(x_i)+\epsilon(Q_C(x_i) + C'(x_i)) \\ \sqrt{\epsilon}R_C(x_i) \end{array}} \boxed{C^{-1}(x_i+\epsilon)} \ldots \boxed{\begin{array}{c} C(x_{i+1})+\epsilon(Q_C(x_{i+1}) + C'(x_{i+1})) \\ \sqrt{\epsilon}R_C(x_{i+1}) \end{array}} \boxed{C^{-1}(x_{i+1}+\epsilon)} \ldots \ . \tag{21}$$

Primes are shorthand for derivatives $\frac{d}{dx}$. The next step is the determination of an update $V(x_i), W(x_i)$ for the matrices $Q_C(x_i), R_C(x_i)$ at each point $x_i$. As in the homogeneous case, we choose as update the local gradient

$$V(x_i) = \frac{\delta}{\delta Q_C^*(x_i)} \frac{\langle\Psi|H|\Psi\rangle}{\langle\Psi|\Psi\rangle}$$
$$W(x_i) = \frac{\delta}{\delta R_C^*(x_i)} \frac{\langle\Psi|H|\Psi\rangle}{\langle\Psi|\Psi\rangle}.$$

For the Hamiltonian Eq.(19), the update $W(x_i)$ to $R_C(x_i)$ has contributions from kinetic energy, potential energy, interaction energy and the environment (see below), respectively. The update $V(x_i)$ to $Q_C(x_i)$ on the other hand has only contributions from kinetic energy and environment. The updates decompose into a sum of the form

$$W(x_i) = W_{kin}(x_i) + W_{pot}(x_i) + W_{int}(x_i) + W_{env}(x_i)$$
$$V(x_i) = V_{kin}(x_i) + V_{env}(x_i).$$

A straight forward but lengthy calculation gives the following results for the contributions to $W(x_i)$ and $V(x_i)$:

$$
\begin{aligned}
W_{env}(x_i) =& H_l(x_i)R_l(x_i)C(x_i) + R_l(x_i)C(x_i)H_r(x_i) \\
W_{pot}(x_i) =& \mu(x_i)R_l(x_i)C(x_i) \\
W_{int}(x_i) =& g\Big( R_l^\dagger(x_i)R_l(x_i)R_l(x_i)C(x_i) + C(x_i)R_r(x_i)R_r(x_i)R_r^\dagger(x_i)|1\rangle\Big) \\
W_{kin}(x_i) =& \frac{1}{2m}\Big( R_l'(x_i)Q_l(x_i)C(x_i) - Q_l'(x_i)R_l(x_i)C(x_i) - Q_l(x_i)R_l'(x_i)C(x_i) - Q_l(x_i)R_l(x_i)C'(x_i) \\
& + R_l(x_i)Q_l'(x_i)C(x_i) + R_l(x_i)Q_l(x_i)C'(x_i) - R_l'(x_i)C'(x_i) - R_l''(x_i)C(x_i) \\
& + Q_l^\dagger(x_i)\big([Q_l(x_i), R_l(x_i)] + R_l'(x_i)\big)C(x_i) - \big([Q_l(x_i), R_l(x_i)] + R_l'(x_i)\big)C(x_i)Q_r^\dagger(x_i)\Big) \\
V_{env}(x_i) =& H_l(x_i)C(x_i) + C(x_i)H_r(x_i) \\
V_{kin}(x_i) =& \frac{1}{2m}\Big( - R_l^\dagger(x_i)\big([Q_l(x_i), R_l(x_i)] + R_l'(x_i)\big)C(x_i) + \big([Q_l(x_i), R_l(x_i)] + R_l'(x_i)\big)C(x_i)R_r^\dagger(x_i)\Big).
\end{aligned}
\tag{22}
$$

Here, $H_l(x_i)$ and $H_r(x_i)$ are hermitian matrices of dimensions $D \times D$ containing the Hamiltonian environment for point $x_i$ (see appendix for details on their calculation). Once the updates have been calculated, the tensors $Q_C^{[m]}(x_i)$ and $R_C^{[m]}(x_i)$ at iteration step $m$ are changed according to

$$
\begin{aligned}
Q_C^{[m]}(x_i) \to & Q^{[m+1]}(x_i) \\
& = [Q_C^{[m]}(x_i) - \alpha V^{[m]}(x_i)][C^{[m]}(x_i)]^{-1} \\
R_C^{[m]}(x_i) \to & R^{[m+1]}(x_i) \\
& = [R_C^{[m]}(x_i) - \alpha W^{[m]}(x_i)][C^{[m]}(x_i)]^{-1}.
\end{aligned}
$$

$\alpha \in \mathbb{R}$ is a small stepsize parameter (typically $\alpha = 10^{-4}\ldots10^{-5}$). From the updated tensors $Q^{[m+1]}(x_i), R^{[m+1]}(x_i)$, a new spMPS is obtained from interpolating the points $(x_i, Q^{[m+1]}(x_i)), (x_i, R^{[m+1]}(x_i))$, which implements a change of the variational parameters,

$$
\begin{aligned}
\mathcal{Q}^{i,[m]} \to & \mathcal{Q}^{i,[m+1]} \\
\mathcal{R}^{i,[m]} \to & \mathcal{R}^{i,[m+1]}.
\end{aligned}
$$

and finishes a single update step. The procedure is then iterated until convergence is reached. As convergence parameter we monitor the norm of the gradient

$$\|\mathcal{G}(x_i)\| = \sqrt{\mathrm{tr}[V(x_i)V^\dagger(x_i) + W(x_i)W^\dagger(x_i)]}. \tag{23}$$

We stop the iteration once $\max[\|\mathcal{G}(x_i)\|] < 10^{-4}$. In our simulations we found that shifting the unit-cell of the system by $L/2$ every $m_{shift} = 5$ update steps has a stabilizing effect on the simulation.

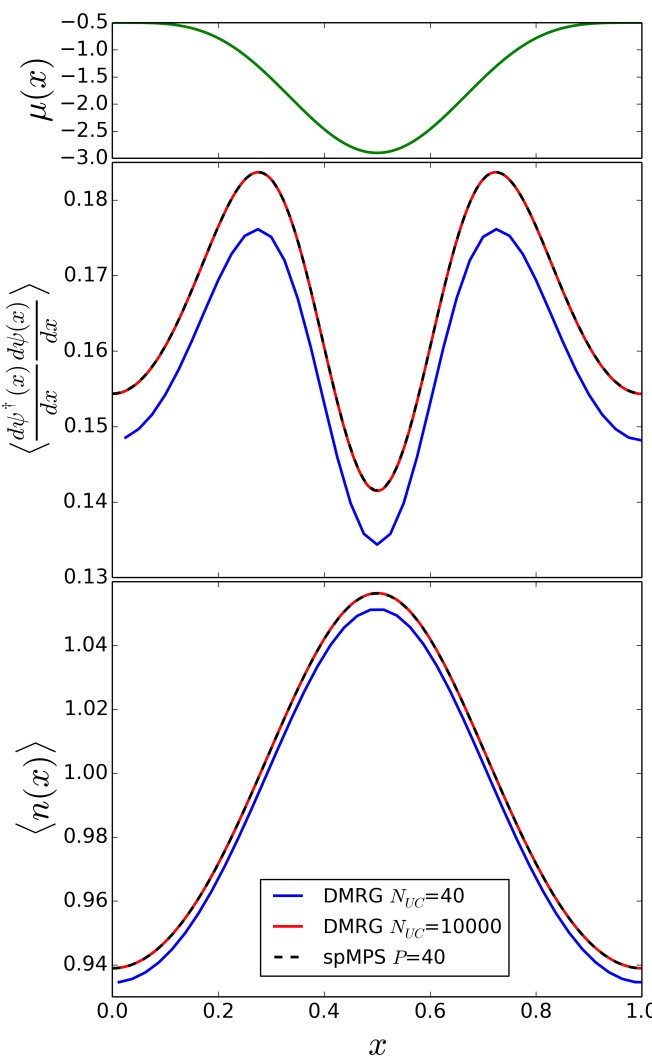

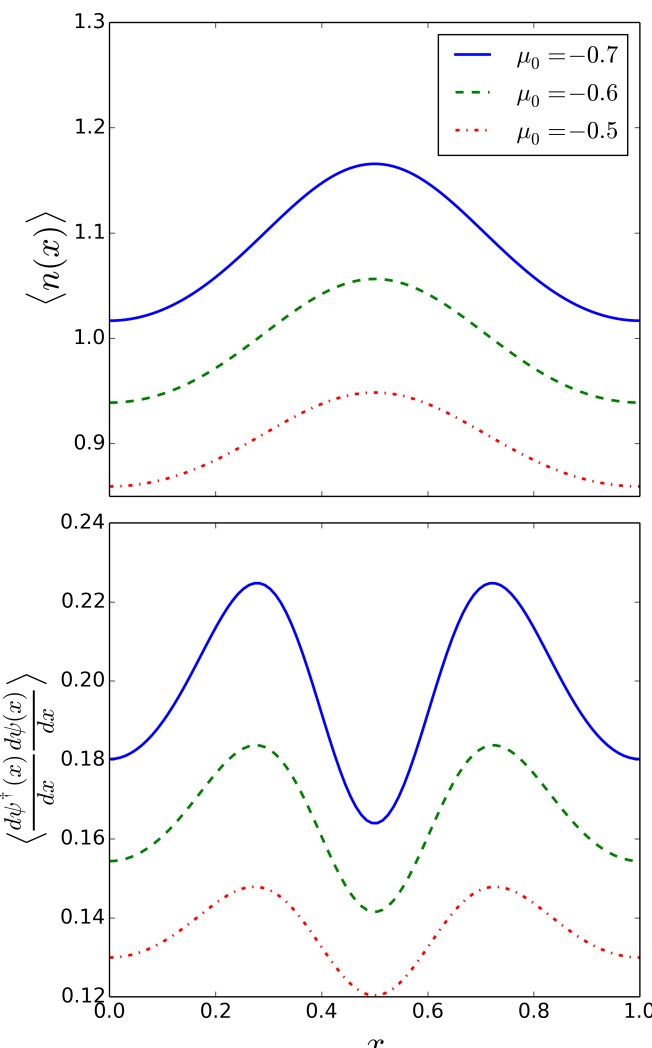

FIG. 2: **Observables in the ground state Eq.(19) for $g = 1.0, m = 0.5, \mu_0 = -0.6$ and a bond dimension $D = 16$.** Top panel: chemical potential $\mu(x)$. We plot kinetic energy density $\langle \frac{d\psi^\dagger(x)}{dx} \frac{d\psi(x)}{dx} \rangle$ (middle panel) and particle density $\langle n(x) \rangle \equiv \langle \psi^\dagger(x)\psi(x) \rangle$ (lower panel) as a function of space-coordinate $x \in [0, L]$ for a single unit-cell. Black dashed lines correspond to spMPS results with $P = 40$ b-spline polynomials. Blue and red solid lines are results from a lattice DMRG calculation, obtained from discretizing the Hamiltonian into $N_{UC} = 40, 10^4$ discretization points per unit-cell, respectively.

## V. EXAMPLE

In the following section we present results for optimized spMPS ground states for a gas of Lieb-Liniger bosons in a periodic potential, see Eq.(19). We emphasize at this point that a good initial state for the spMPS optimization is important for a stable ground state optimization. The main problem we encounter in our simulations are diverging matrix functions $Q(x), R(x)$ with ongoing optimization. Usually, these divergences can have multiple

FIG. 3: **Ground state observables for different amplitudes of the chemical potential.** Particle density $\langle n(x) \rangle$ (upper panel) and kinetic energy density (lower panel) for different amplitudes $\mu_0 = -0.5, -0.6, -0.7$ and $g = 1.0, m = 0.5, D = 16$.

causes, and it is not always possible to pin down the exact cause for any particular case. One well known problem is related to the step size $\alpha$ in the optimization and the average distance $\Delta x = x_i - x_{i-1}$ between two interpolation points. The so-called Courant-Levy condition determines a maximally possible step size $\alpha_{max}$ for a given $\Delta x$ before the iteration diverges. Additionally to this, we observe that for large gradients, the step size has to be taken very small in order to avoid any divergences. For random initial states, the gradients will typically become so large that the step sizes get too small to reach convergence within any reasonable time frame. Starting from a good initial guess for a spMPS wave function reduces many of these problems considerably and leads to a sizeable reduction of computational run times. In this manuscript we use an optimization and interpolation method for lattice MPS to get a good initial guess for

the spMPS optimization, which we then further optimize with our proposed optimization method. The details of this initialization method will be published elsewhere[41].

In Fig. 2 we show results for the kinetic energy density $\langle \frac{d\psi^\dagger}{dx} \frac{d\psi}{dx} \rangle$ (middle panel) and particle $\langle n(x) \rangle$ (lower panel) for an approximate ground state of Eq.(19) as obtained by the proposed spMPS optimization (black dashed line), for a bond dimension $D = 16$. The top panel shows the chemical potential $\mu(x)$ Eq.(20) for $\mu_0 = -0.6$. We used a spMPS of order $k = 5$ and $P = 40$ interpolation polynomials $B_i^k(x)$ within a unit-cell. The observables are exactly periodic with the same periodicity as the chemical potential . For comparison, we show results from a DMRG calculation, with a discretization of Eq.(19) using $N_{UC} = 40$ discretization points in the unit-cell (blue solid line). As reference, we also show a DMRG calculation using $N_{UC} = 10^4$ discretization points per unit-cell, which we consider to be the variational optimum for a chosen bond dimension $D = 16$, and to be practically free of discretization artifacts. The exceptional agreement of the spMPS results with the $N_{UC} = 10^4$ DMRG calculation is evidence that the spMPS wave function is essentially free of any discretization artifacts.

While the spMPS wavefunction achieves high accuracy with a small number of variational parameters, and the computational cost of the spMPS optimization has formally a scaling with bond dimension identical to DMRG ($D^3$), the DMRG, even on very fine grids, is currently still faster to optimize. This is mainly due to the fact that at each step in the spMPS optimization, calculation of so called left and right reduced density matrices is done using high precision ODE solvers in scipy, which usually is the most time consuming step. This can be overcome e.g. by projecting the transfer operator into the basis-spline space and solving for the left and right dominant eigenvectors directly, which will be part of a future publication.

In Fig. 3 we show results for particle density $\langle n(x) \rangle$ and kinetic energy density $\langle \frac{d\psi^\dagger}{dx} \frac{d\psi}{dx} \rangle$ for different values of the amplitude $\mu_0$ of the chemical potential (see Eq.(20)). As expected, raising the amplitude causes a similar increase in the amplitudes for the particle density, as well as the kinetic energy density.

## VI. CONCLUSIONS AND OUTLOOK

We have proposed a novel parametrization for non-translational invariant continuous Matrix Product States in terms of a b-spline interpolation for the cMPS matrix-functions $Q(x), R(x)$. The b-spline parametrization allows for an efficient representation of inhomogeneous continuous many-body wave functions, and is free of any discretization artifacts. We extend well known regauging techniques for lattice Matrix Product States to the case of spline-based Matrix Product States (spMPS), and show how a recently developed optimization method for translational invariant cMPS can be applied to the case

of spMPS. We apply the method to a gas of interacting Lieb-Liniger bosons in a periodic potential, where a comparison with high-precision Density Matrix Renormalization Group calculations underpins that the proposed variational class of spMPS is free of any discretization errors. It is thus well suited to parametrize ground states of continuous quantum field theories without translational invariance.

The optimization method which we have proposed in this paper uses a point-wise update of the spMPS tensors $Q(x_i), R(x_i)$ at points $x_i$, followed by an interpolation to new matrix-functions $\tilde{Q}(x), \tilde{R}(x)$ using the points $(x_i, Q(x_i)), (x_i, R(x_i))$ as interpolation points. The variational parameters $\mathcal{Q}^i, \mathcal{R}^i$ get updated via the interpolation step. It would be interesting to see if the tensors $\mathcal{Q}^i, \mathcal{R}^i$ could be updated directly as well. For example, for a given $i$, a variation of the tensors $\mathcal{Q}^i, \mathcal{R}^i$ introduces local changes of the tensors $Q(x), R(x)$, where the changes are localized around position $x_i$, and are smooth in space, due to the smoothness and locality of the b-spline polynomials $B_i^k(x)$. This local change gives rise to a change in the energy expectation value and, if chosen appropriately, will lower the energy. To find an appropriate variation, one could for example linearize the energy functional around the current values of $\mathcal{Q}^i, \mathcal{R}^i$. Such an approach could potentially considerably enhance the speed of convergence for ground state optimizations for spMPS.

Two other important extensions of spMPS are systems with multiple species of bosons or fermions, and systems with open boundaries. In first case, each species $\alpha$ gives rise to a matrix $R_\alpha(x)$. The b-spline interpolation can be applied to each $R_\alpha(x)$ individually. To ensure the necessary regularity conditions[34], suitable parametrizations for the matrices $R_\alpha(x)$ can be combined with the b-spline interpolation. The case of open boundary conditions is more delicate. Current cMPS approaches to such systems suffer from rank deficiency problems at the boundaries. These problems are expected to persist in the case of a spMPS parametrization. However, due to the local nature of the basis splines, the spMPS approach could be combined with regular lattice aproaches close to the boundaries of the system[42]. For example, one could use lattice MPS methods very close to the boundaries, and a spMPS parametrization in the bulk. The two parametrizations could then be patched together to circumvent the rank deficiency problem

Finally, the ability to use a continuous-space parametrization of the many-body wave function $|\Psi\rangle$ opens the possibility to combine tools for solving partial differential equations (PDEs) developed in numerical mathematics to be combined with non-perturbative tensor network approaches to interacting quantum field theories. Vice versa, the methods developed here may have important applications in the field of numerical mathematics of coupled, non-linear PDE.

## VII. ACKNOWLEDGEMENTS

The author thanks J. Rincón, A. Milsted and G. Vidal for useful discussions and G. Vidal for valuable comments on the manuscript. The author also acknowledges support by the Simons Foundation (Many Electron Collaboration). Computations were made on the supercomputer Mammouth parallèle II from University of Sherbrooke, managed by Calcul Québec and Compute Canada. The operation of this supercomputer is funded by the Canada Foundation for Innovation (CFI), the ministère de l'Économie, de la science et de l'innovation du Québec (MESI) and the Fonds de recherche du Québec - Nature et technologies (FRQ-NT). This research was supported in part by Perimeter Institute for Theoretical Physics. Research at Perimeter Institute is supported by the Government of Canada through Industry Canada and by the Province of Ontario through the Ministry of Economic Development & Innovation.

* Electronic address: martin.ganahl@gmail.com

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

## VIII. APPENDIX

### A. Obtaining the central canonical form of a cMPS

The calculation for obtaining the central canonical form proceeds largely along the same lines as for the left-orthogonal gauge. The first step is the determination of the left and right steady-state reduced den-

sity matrices $\langle l(x)|, |r(x)\rangle$. On each link between positions $x$ and $x + \epsilon$, (with a tensor $A^\sigma(x)$ in between) we insert resolutions of the identity of the form

$$\left[\sqrt{l(x)}\right]^{-1}\sqrt{l(x)}\sqrt{r(x)}\left[\sqrt{r(x)}\right]^{-1}:$$

$$\begin{pmatrix} 1 + \epsilon Q(x) \\ \sqrt{\epsilon}R(x) \end{pmatrix} = \left[\sqrt{l(x)}\right]^{-1}\sqrt{l(x)}\sqrt{r(x)}\left[\sqrt{r(x)}\right]^{-1}\begin{pmatrix} 1 + \epsilon Q(x) \\ \sqrt{\epsilon}R(x) \end{pmatrix}\left[\sqrt{l(x+\epsilon)}\right]^{-1}\sqrt{l(x+\epsilon)}\sqrt{r(x+\epsilon)}\left[\sqrt{r(x+\epsilon)}\right]^{-1}. \tag{A1}$$

With the definition

$$C(x) \equiv \sqrt{l(x)}\sqrt{r(x)}$$

and the expansion

$$\frac{1}{\sqrt{l(x+\epsilon)}} = \frac{1}{\sqrt{l(x)}}\left(1 - \epsilon\frac{d\sqrt{l(x)}}{dx}\left[\sqrt{l(x)}\right]^{-1}\right) + \mathcal{O}(\epsilon^2),$$

a simple calculations turns Eq.(A1) into

$$\begin{pmatrix} 1 + \epsilon Q(x) \\ \sqrt{\epsilon}R(x) \end{pmatrix} = \left[\sqrt{l(x)}\right]^{-1}\underbrace{C(x)\frac{1}{\sqrt{r(x)}}\begin{pmatrix} 1 + \epsilon Q(x) \\ \sqrt{\epsilon}R(x) \end{pmatrix}\frac{1}{\sqrt{l(x)}}\left(1 - \epsilon\frac{d\sqrt{l(x)}}{dx}\frac{1}{\sqrt{l(x)}}\right)\left(C(x) + \epsilon\frac{dC}{dx}\right)}_{\text{left normalized}}\left[\sqrt{r(x+\epsilon)}\right]^{-1}$$

$$= \left[\sqrt{l(x)}\right]^{-1}C(x)\underbrace{\frac{1}{\sqrt{r(x)}}\begin{pmatrix} 1 + \epsilon Q(x) \\ \sqrt{\epsilon}R(x) \end{pmatrix}\frac{1}{\sqrt{l(x)}}\left(1 - \epsilon\frac{d\sqrt{l(x)}}{dx}\frac{1}{\sqrt{l(x)}}\right)\left(C(x) + \epsilon\frac{dC}{dx}\right)\left[\sqrt{r(x+\epsilon)}\right]^{-1}}_{\text{right normalized}},$$

where we have dropped terms of order $\mathcal{O}(\epsilon^2)$, and we have indicated how to contract matrices such that a left or right normalized cMPS is obtained. The central canonical form is defined by matrices $\Gamma_Q(x), \Gamma_R(x), C(x)$ and $\frac{dC(x)}{dx}$, such that

$$Q_l = C(x)\Gamma_Q(x)$$
$$R_l = C(x)\Gamma_R(x)$$
$$Q_r = \Gamma_Q(x)C(x) + [C(x)]^{-1}\frac{dC}{dx}$$
$$R_r = \Gamma_R(x)C(x).$$

It is then just a matter of collecting orders of $\epsilon$ to see that the matrices

$$\Gamma_Q(x) = \frac{1}{\sqrt{r(x)}}Q(x)\frac{1}{\sqrt{l(x)}}$$
$$- \frac{1}{\sqrt{r(x)}}\frac{1}{\sqrt{l(x)}}\frac{d\sqrt{l(x)}}{dx}\frac{1}{\sqrt{l(x)}}$$
$$\Gamma_R(x) = \frac{1}{\sqrt{r(x)}}R(x)\frac{1}{\sqrt{l(x)}}$$

fullfill this condition. Note that when permuting $C(x)$ past $Q_l(x)$ or $Q_r(x)$, we have to use

$$C(x)Q_r(x) = Q_l(x)C(x) + \frac{dC}{dx}. \tag{A2}$$

### B. Calculation of environmental contributions

In this section we will outline the calculation of the matrices $H_l(x_i), H_r(x_i)$ encountered in Eq.(22). In DMRG parlance, these hermitian matrices are the left and right Hamiltonian environments of a local site at position $x_i$. For example, $H_l(x_i)$ contains all contributions $\int_{-\infty}^{x_i} h(x)dx$ to the left of $x_i$. On a more technical level, $H_l$ and $H_r$ are the projections of $\int_{-\infty}^{x_i} h(x)dx$ and $\int_{x_i}^{\infty} h(x)dx$ into two reduced orthonormal basis sets at position $x_i$. At this point it is helpful to introduce the standard MPS diagramatic technique to visualize the following calculations. To this end we use the diagram

$$\ldots - \boxed{x_i - \epsilon} - \boxed{x_i} - \boxed{x_i + \epsilon} - \ldots \tag{A3}$$

to denote an MPO representation of a discretization of the Hamiltonian Eq.(19), with lattice spacing $\epsilon$. Note that the following diagrams serve only as a visual aid, and network contractions are performed solely by integration of a system of ordinary differential equations (see below). In this diagrammatic notation $H_l(x_i)$ and $H_r(x_i)$ are given by half-infinite tensor networks of the form

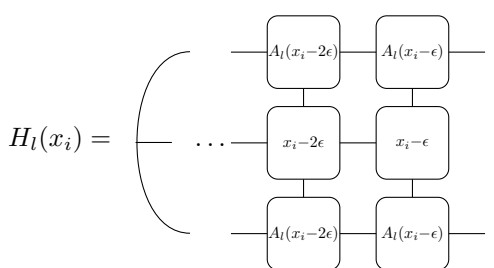

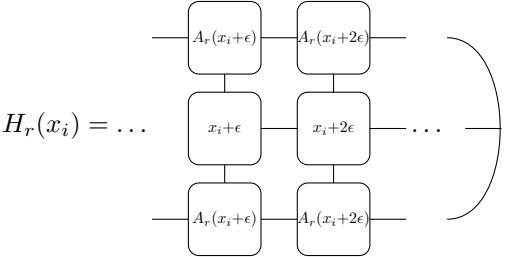

$\big(\!\!\text{ and }\!\!\big)$ are arbitrary tensors at $x = \pm\infty$. Due to the extensivity of energy $H_l(x_i)$ and $H_r(x_i)$ both contain infinities which, for numerical computations, have to be regularized[31], as will be detailed in the following. To ease up notation we use the abbrevation

$$\mathcal{T}_{l/r}(x_i) \equiv \mathcal{P}e^{\int_{x_i}^{x_i+L} T_{l/r}(x)dx}$$
$$T_{l/r}(x) = Q_{l/r}(x) \otimes \mathbb{1} + \mathbb{1} \otimes Q^*_{l/r}(x) + R_{l/r}(x) \otimes R^*_{l/r}(x)$$

in the following. For a periodic system, the calculation of $\langle H_l(x_i)|$ and $|H_r(x_i)\rangle$ can be simplified to

$$\langle H_l(x_i)| = \langle h_l(x_i)| \sum_{n\in\mathbb{N}} [\mathcal{T}_l(x_i)]^n \tag{A4}$$

$$|H_r(x_i)\rangle = \sum_{n\in\mathbb{N}} [\mathcal{T}_r(x_i)]^n |h_r(x_i)\rangle. \tag{A5}$$

The hermitian matrices $\langle h_l(x_i)|$ and $|h_r(x_i)\rangle$ are defined as the contraction of an infinite tensor network of length $L$ of the form

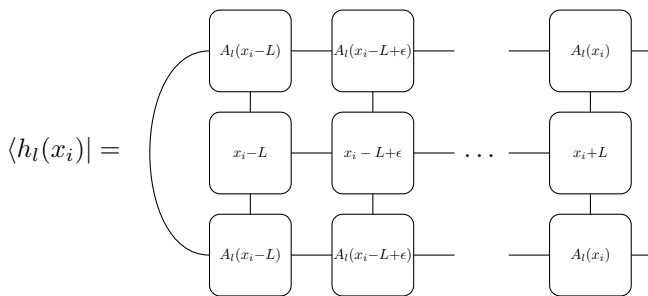

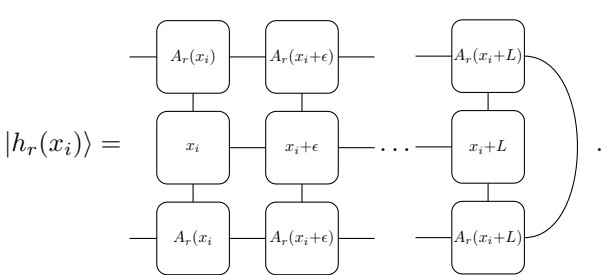

They are called the effective unit-cell Hamiltonians, i.e. the Hamiltonian operator of a unit-cell projected into an effective, orthonormal basis set of the left respectively right infinite half-chain (given by left and right isometric tensors $A^\sigma_{l/r}(x)$). The contraction of these networks is equivalent to solving a boundary-value problem for a system of ordinary differential equations for the matrices $\langle h_l(x)|$ and $|h_r(x)\rangle$, in analogy to the evolution of e.g. $\langle l(x)|$ with the transfer operator $\mathcal{T}(x,y)$. In semigraphical notation, these differential equations are given by

$$\frac{d}{dx}\langle h_l(x)| = \frac{1}{2m}\left(\frac{[Q_l(x),R_l(x)]+\frac{dR_l(x)}{dx}}{[Q_l(x)^*,R_l(x)^*]+\frac{dR_l(x)^*}{dx}}\right) + g\left(\frac{R_l^2(x)}{R_l^{*2}(x)}\right) + \mu(x)\left(\frac{R_l(x)}{R_l^*(x)}\right) + \langle h_l(x)|T_l(x) \tag{A6}$$

$$\frac{d}{dx}|h_r(x)\rangle = \frac{1}{2m}\left(\frac{[Q_r(x),R_r(x)]+\frac{dR_r(x)}{dx}}{[Q_r(x)^*,R_r(x)^*]+\frac{dR_r(x)^*}{dx}}\right) + g\left(\frac{R_r^2(x)}{R_r^{*2}(x)}\right) + \mu(x)\left(\frac{R_r(x)}{R_r^*(x)}\right) + T_r(x)|h_r(x)\rangle \tag{A7}$$

with $T_r |h_r(x)\rangle$ and $\langle h_l(x)| T_l$ given by Eq.(9) and Eq.(10), and boundary conditions

$$\langle h_l(x_i - L)| = 0$$
$$|h_r(x_i + L)\rangle = 0.$$

$\langle h_l(x_i)|$ $(|h_r(x_i)\rangle)$ is then obtained by evolving $\langle h_l(x_i - L)|$ $(|h_r(x_i + L)\rangle)$ from $x = x_i - L$ $(x = x_i + L)$ to $x = x_i$.

After proper normalization of the spMPS, the unit-cell transfer operator $\mathcal{T}_l(x_i)$ has a left eigenmatrix $\langle \mathbb{1}|$ with eigenvalue $\eta = 1$, and similarly $\mathcal{T}_r(x_i)$ has a right eigenmatrix $|\mathbb{1}\rangle$ with the same eigenvalue $\eta = 1$. The geometric series Eq.(A4) and Eq.(A5) thus diverge and the sums cannot be performed trivially. The infinities in Eq.(A4) and Eq.(A5) are due to an infinite energy expectation value of the cMPS $|\Psi\rangle$. Since this is just an infinite energy offset, one can savely subtract it from the Hamiltonian. One way of achieving this is by a proper regularization of the unit-cell transfer operator $\mathcal{T}_{l/r}(x_i)^{43}$. This is done by projecting it into the subspace orthogonal to $|\mathbb{1}\rangle \langle l(x_i)|$ and $|r(x_i)\rangle \langle \mathbb{1}|$, respectively, i.e. by replacing $\mathcal{T}_{l/r}(x_i)$ with

$$\mathcal{T}_{l\perp}(x_i) = \mathcal{T}_l(x_i) - |r(x_i)\rangle \langle \mathbb{1}|$$
$$\mathcal{T}_{r\perp}(x_i) = \mathcal{T}_r(x_i) - |\mathbb{1}\rangle \langle l(x_i)|$$
$$\langle h_l(x_i)|_\perp = \langle h_l(x_i)| - \langle h_l(x_i)|r(x_i)\rangle \langle \mathbb{1}|$$
$$|h_r(x_i)\rangle_\perp = |h_r(x_i)\rangle - \langle l(x_i)|h_r(x_i)\rangle |\mathbb{1}\rangle .$$

We then solve for $\langle H_l(x_i)|$ and $|H_r(x_i)\rangle$ by inverting the equations

$$\langle h_l(x_i)|_\perp \frac{1}{\mathbb{1} - \mathcal{T}_{l\perp}(x_i)} = \langle H_l(x_i)| \qquad (A8)$$

$$\frac{1}{\mathbb{1} - \mathcal{T}_{r\perp}(x_i)} |h_r(x_i)\rangle_\perp = |H_r(x_i)\rangle \qquad (A9)$$

using a sparse solver, like e.g. the *lgmres* routine provided by the *scientific python* package. In practice it is not necessary to solve Eq.(A8) and Eq.(A9) for each $x_i$. Instead, we first solve it for a single point, e.g. $x_i = 0$, and then evolve $\langle H_l(0)|$ and $|H_r(0)\rangle$ to all other points $x_i$ using Eq.(A6) and Eq.(A7).