# Peer review of "Continuous Matrix Product States for Inhomogeneous Quantum Field Theories: a Basis-Spline Approach"

_SciPost Physics_

## Round 1 · Referee Report · Anonymous (Referee 1) · 2019-4-3

Strengths

1- Clear and concise 2- Proposes a technical implementation that may boost continusous-space MPS 3- Timely and potentially important for the field

Weaknesses

1- The comparison to algorithms based on naive discretization is very disappointing 2- Not clear whether the approach can be useful in practice or not 3- Benchmarking based on a single example

Report

The paper proposes a new implementation of matrix product state (MPS) - based tensor network techniques for continuous-space Hamiltonians. It builds upon an approach co-authored by the author of this contribution and introduces a spline parametrization of the MPS wave function. This is a standard approach, which usually allows for a better representation of continuous-space functions, as compared to naive, homogeneous discretization.

The paper is well written and easy to read. It contains all the pieces of information necessary to follow the approach, without excessive details. The limitations of the approach are discussed on the basis of calculations of the ground state of one-dimensional bosons in a periodic lattice.

While the approach is a priori well justified and sound, it is far not clear that it provides any advantage in practice. This is a central point because the main contribution here is a rather technical implementation. The results presented in the paper are actually very diapointing and the conclusion is that it is better, in practice, to use a standard naive discretization. This is the main flaw of the paper. The author should provide convincing arguments that the method could be beneficial, maybe using different examples of physical systems to be solved or implementing the proposed improvement.

Alternatively, the abstract and the introduction should fairly state that the present implementation suffers strong limitations, but, possibly, that ways to overcome the latter are discussed. In the present form, I think that the sentence "We will demonstrate the power of the spMPS ansatz by obtaining the ground state of a gas of interacting Lieb-Liniger bosons in a periodic potential." is not a fair account of the results presented in the paper.

Requested changes

  • The generic form in Eq. (1) should be justified
  • The definition of the A^\sigma in terms of product matrices should be related to the wavefunction at the very beginning of Sec. III, where they are refered to
  • The notations in the formulas for A^0 and A^\sigma are a bit misleading: Why is R(x) inside brackets, Q(x) is not?
  • "For the cases considered in this manuscript, tensors A^\sigma with \sigma > 1 can be neglected." Why?
  • Typo "x1" on the very first line
  • The notion of "benchmarking" may be better stressed at the end of the introduction
  • "an SVD" -> "a SVD"
  • Why is the form of the potential in Eq. (20) so complicated, and not just a sine function?
  • "particle \langle n(x) \rangle" -> "particle density \langle n(x) \rangle"

---

## Round 1 · Referee Report · Anonymous (Referee 2) · 2019-4-26

Strengths

1-detailed description of the methodology (i.e. the algorithm used, and specific routines used in Python)
2-first demonstration of continuous matrix product states in an inhomogeneous setting
3-fair discussion of limitations of the current approach

Weaknesses

1-technique that this paper introduces is only illustrated on/applied to a single example
2-specific quantitative details about the example/results are vague or missing: i.e what are the runtimes of the variational spMPS versus the reference DMRG, how many variational parameters has the former versus the latter, ...
3-introduction to splines is not particularly enlightning

Report

In this manuscript, the author applies the continuous matrix product state ansatz of Verstraete and Cirac to inhomogeneous quantum field theories. So far, their use had been restricted to translation-invariant systems, in which case the variational parameters correspond to a single set of matrices. For an inhomogenous system, the variational parameters are instead matrix-valued functions of space, and a particular way needs to be chosen to represent those using a finite number of parameters stored on a computer. Here, the authors choose to combine spatial discretisation and spline interpolation.

The technical aspects of this approach and its implementation are described in a self-contained way, without becoming too elaborate. The only subsection of Sections II to IV which I found to be of lesser quality is the introduction to splines. For example, the knot points are introduced, but their role or use is not really discussed (especially when they are degenerate). Furthermore, most of this section is about splines for open boundary conditions, whereas the author then uses splines with periodic boundary conditions.

The example (Section V) is somewhat weak. Only a single example is studied (and to know which potential this exactly represents, one has to browse back two pages, it would be better stated in Section V). Furthermore, this section provides very little quantitative details and leaves the reader with many questions, such as * How much do the DMRG reference result (N=10^4) and the spMPS result differ, e.g. for the particle density, or for the total energy in one unit cell? * What are the actual runtimes of the algorithms, the number of optimization steps? * Does it work better or worse at larger bond dimensions? Is there a big improvement over Gross-Pitaevskii (D=1)? Are there specific effect the latter cannot capture? * How much variational parameters do the spMPS and the DMRG/MPS ansatz contain? * There is a big gap between N=40 and N=10^4, how would DMRG do with N = 400, 1000, 4000? * Is there any limit in which comparison with an exact result or some other technique is possible?

Other comments and typos: * Both in Section III and in the appendix, the central canonical form is introduced, with matrices (matrix functions) Gamma_Q(x) and Gamma_R(x). However, these do not seem to be used in the actual algorithm, only Q_l, Q_C, Q_r, R_l, R_C, R_r and C are used. What is the relevance of the Gamma_Q and Gamma_R or why are they also introduced?

  • Abstract: "We show that spMPS achieve a reduction of variational parameters by two orders of magnitude...". Firstly, these numbers should be made more explicit in the Example section, as stated above. Secondly, the abstract should also point out the limitations, i.e. that optimization of this state is still hard, slow and unstable.

  • "x1computational" in the first sentence

  • "savely" -> "safely" in the last paragraph of the appendix

Requested changes

  • Improve the example section, addressing the questions posed above.
  • Address other comments

---

## Editorial Decision

awaiting_resubmission